# Barriers and Facilitators of Ambient Assisted Living Systems: A Systematic Literature Review

**DOI:** 10.3390/ijerph20065020

**Published:** 2023-03-12

**Authors:** Gastón Márquez, Carla Taramasco

**Affiliations:** 1Departamento de Electrónica e Informática, Universidad Técnica Federico Santa María, Millennium Nucleus on Sociomedicine, Concepción 4030000, Chile; 2Facultad de Ingeniería, Universidad Andrés Bello, Millennium Nucleus on Sociomedicine, Viña del Mar 2520000, Chile

**Keywords:** Ambient Assisted Living Systems, barriers, facilitators

## Abstract

Ambient Assisted Living Systems (AALSs) use information and communication technologies to support care for the growing population of older adults. AALSs focus on providing multidimensional support to families, primary care facilities, and patients to improve the quality of life of the elderly. The literature has studied the qualities of AALSs from different perspectives; however, there has been little discussion regarding the operational experience of developing and deploying such systems. This paper presents a literature review based on the PRISMA methodology regarding operational facilitators and barriers of AALSs. This study identified 750 papers, of which 61 were selected. The results indicated that the selected studies mentioned more barriers than facilitators. Both barriers and facilitators concentrate on aspects of developing and configuring the technological infrastructure of AALSs. This study organizes and describes the current literature on the challenges and opportunities regarding the operation of AALSs in practice, which translates into support for practitioners when developing and deploying AALSs.

## 1. Introduction

According to the World Health Organization [1], every country is experiencing an increase in the number and proportion of older and disabled people. By 2030, one in six people will be aged 60 years or older. At that time, the population aged 60 years and over will have risen from 1 billion in 2020 to 1.4 billion. Moreover, by 2050, the world’s population of people aged 60 and over will double (2.1 billion). In this scenario, governments worldwide have been implementing different public policies focused on the care and welfare of the elderly. Nevertheless, although some governments have promoted protection policies, they have not been effective because their families perceive them as a responsibility that is difficult to address [2]. In addition, they require special time and care because their physical and cognitive conditions deteriorate due to the aging process, which makes them dependent on their families and society.

The proposed solutions to this problem require the assistance of medical personnel and specialists in the care of older and disabled people [3]. Nevertheless, these solutions are generally costly, and there are not enough staff to keep up with everything that happens to the elderly. Therefore, it is necessary to address other alternatives that allow older adults to be assisted from home simply and practically, which do not require significant effort in mobility and learning. In this scenario, Ambient Assisted Living Systems (AALSs) have emerged as an alternative [4]. AALSs deal with the integration of sensors and intelligent devices that can be portable and environmental, and they send information to process and store data to analyze physical and comfort parameters to enable readjustment according to the conditions of older adults with disabilities. The relevance of these systems lies in the fact that they are concerned with the prevention and care of the elderly both at home and outside the home during emergencies [5].

The academic literature has studied AALSs from different points of view, such as architecture [6,7], security [8], and opportunities [4]. Several studies have proposed sophisticated techniques for the prevention and surveillance of disabilities in older adults and disabled people [9,10,11]. However, there has been little discussion at the operational level regarding the performance of AALSs. Although current studies have been significant and relevant in describing AALS proposals and solutions, they are impractical for explaining the benefits and difficulties of implementing AALS at the system and patient levels. This situation creates a knowledge gap for practitioners and clinicians regarding AALSs in the real-world setting.

This paper describes operational facilitators and barriers related to the development and deployment of AALS. Through a systematic review of the literature based on the PRISMA methodology [12], we analyzed 61 primary studies to identify facilitators and barriers from an operational point of view. We classified the studies based on the architectural views proposed by Kruchten [13], which describe a system based on multiple viewpoints. The main contribution of this study is a pragmatic classification of barriers and facilitators regarding the use of AALSs from a system and patient point of view. In addition, the results of our study can be used as a starting point for practitioners who require the implementation and deployment of AALSs.

The structure of this paper is described as follows: Section 2 introduces AALSs; Section 3 details related work; Section 4 details the study search protocol; Section 5 describes the results; Section 6 discusses key findings; Section 7 discusses the limitations of our study; and Section 8 describes the conclusions.

## 2. Ambient Assisted Living Systems

AALSs are a set of technologies whose main objective is to increase the time people spend living autonomously [14]. Such systems are primarily aimed at older adults, disabled people, or caregivers, with the idea that they can be extrapolated to other social groups such as people with functional diversity. The technologies, programs, and applications that encompass AALSs must be able to collect data but also monitor (and try to correct, as far as possible) the life habits of the users [5].

AALSs use devices to help older people and disabled people clean, cook, shop, take care of hygiene, diet and monitor medication and health care. Beyond support for housework and personal care, these devices have also helped prevent injuries at home and alert support services in case of a fall, to facilitate people’s mobility and ensure that they can move around on their own, or to foster their social integration and reduce feelings of loneliness and isolation [11]. The technologies used in AALSs have been designed to enable the elderly and people with functional diversity to live independently, helping them perform everyday tasks that they usually cannot perform on their own. In this way, an environment in which people can continue to be self-sufficient citizens is created.

Figure 1 describes the basic architecture of an AALS [15]. Both sensors and medical devices act as inputs to the system, which, through a communication channel, send their data to a component whose functionality is to collect the obtained data. In this component, pre-processing algorithms are executed that clean, load, and transform the data obtained from the sources to obtain structured and ordered data. Using another communication channel, the data can be sent to a component that manages the data to be prepared for end users. In parallel, it is possible to visualize data that has already been processed using the same communication channel. Finally, once the data are managed, they can be sent to different endpoints for various purposes. These purposes can be the visualization of patients by relatives and care centers or storage in the cloud.

## 3. Related Work

Sun et al. [4] discuss current issues regarding the development of AAL systems for older people. The authors describe many efforts to build home care environments, but significant challenges need to be addressed. For example, one of the challenges identified is the potential for social isolation due to the overuse of technology and the lack of communication between assisted persons and the outside community. On the other hand, the authors identified that governments are making more efforts to establish social connections between assisted persons and the outside world.

Garcés et al. [6] describe models and quality attributes related to AALSs. The authors identified the main quality attributes that identify AALSs, how the attributes were defined and assessed, and the subdomains of AALSs where they were proposed. The results of this study indicate that there is still a need for greater industry involvement in the engineering of AALSs to establish quality management and its associated quality assurance, which could be considered essential for any AALS system.

Abtoy et al. [7] discuss the concepts of reference models and architectures for designing AALSs. The authors describe different architectural styles to summarize the weaknesses of each. In turn, the authors proposed a set of dimensions and classifications to highlight the main areas that need to be addressed by AALSs. The authors highlight the need for ease, interoperability, and independence in AALSs.

Choukou et al. [16] conduct a systematic review of the literature aimed at scoping and reporting advances in AALSs in terms of health outcomes in older adults. The authors described that AALSs made a significant contribution to patient quality of life. However, they also mentioned that there is insufficient empirical evidence regarding AALS and patient care.

Dimitrievski et al. [14] mention that current research on AALSs lacks sufficient experimental results with data from continuous monitoring. In turn, the authors describe that AALSs is a field that has the potential to benefit from machine learning based on the data that is processed by devices and sensors. In turn, the authors mention that to create a supportive environment that does not stigmatize older people, the focus should be on collecting data from non-intrusive environmental sensors.

Jovanovic et al. [17] conduct a review of the literature on artificial intelligence models in AALSs. The authors discussed specific artificial intelligence models that are widely used in AALSs. In turn, the authors focused on articulating research toward end-users, health-care professionals, researchers, and practitioners in the development of intelligent AALSs.

Cicirelli et al. [18] analyze the AALSs literature from the point of view of existing contexts, technologies and approaches to characterize AALSs development needs. The authors’ comprehensive study focuses on AALSs’ contexts, potential users, and functionalities as well as the technologies and methodologies used for AALSs’ development and deployment.

The studies described in this section make significant contributions to research related to AALSs. There are different alternatives to how AALSs contribute to improving the quality of life of older adult and disabled patients. However, to the best of our knowledge, related work has been limited to discussing the benefits and challenges that AALSs present in real-world and operational settings. Therefore, our study aims to address this need for more knowledge by focusing on a systematic review of a more practical discussion of using AALSs based on facilitators and barriers.

## 4. Methods

Our study aims to identify and describe the main operational and technical barriers and facilitators of AALSs in order to provide a body of knowledge to derive new solutions and guidelines for implementing and using AALSs. We used the PRISMA methodology [12] to identify and select primary studies based on three steps: identification, screening and included. The main feature of PRISMA is that it provides items that allow reporting in literature reviews to perform data analysis and meta-analysis. Therefore, the research questions are as follows:Which are the main operational barriers to AALSs?Which are the main operational facilitators to AALSs?

### 4.1. Identification

In this step, we identified the keywords based on the population, intervention, comparison and output approaches [19] (see Table 1) in order to define a search string according to the objective of our research With regard to comparison, this approach cannot be applied because there are no previous studies on our goal that can serve as a basis for comparison. Subsequently, we joined the identified keywords using the “AND” and “OR” operators to obtain the search string. With the search string created, we proceed to search for primary studies in the following databases: Web of Science (https://clarivate.com/webofsciencegroup, accessed on January 2022), PubMed (https://pubmed.ncbi.nlm.nih.gov, accessed on January 2022), ACM Digital Library (https://dl.acm.org, accessed on January 2022), ScienceDirect (https://www.sciencedirect.com, accessed on March 2022), IEEE Xplorer (https://ieeexplore.ieee.org/Xplore/home.jsp, accessed on June 2022) and Wiley (https://onlinelibrary.wiley.com, accessed on October 2022). The identification process was conducted between January and October 2022. This process involved the study’s principal researchers and two collaborators, which were health professionals with knowledge of caring for the elderly and people with disabilities. Through collaborative work, we discussed each piece of information extracted from primary studies based on these views. In this way, we were able to refine the results of our study and thus decrease bias in the results.

### 4.2. Screening

We analyzed and refined the primary studies using the inclusion and exclusion criteria. Both the criteria are described as follows:Inclusion criteria
–Papers must be related to health and AALSs.–Papers must be able to describe the results of their studies explicitly.–Papers must detail technological aspects of AALSs.–Papers discuss the advantages or disadvantages of AALSs.–Papers report, as far as possible, the experience of using AALSs.Exclusion criteria–The paper does not detail technical or methodological aspects of AALSs.–There is no evidence of the advantages or disadvantages of AALSs.–The paper does not describe the results of your research.–The paper is less than four pages long.

To further expand our search for primary studies, we applied the snowball method [20]. This method is a sampling method used when information from primary studies is difficult to find. For our study, we used this method in order to increase the scope of the search for primary studies. Using Google Scholar (https://scholar.google.com, accessed on October 2022), we used backward and forward snowballing methods (i.e., references and citations).

### 4.3. Included

In this step, we obtain the final set of primary studies. To organize this information, we created a template (see Table 2) to detail each primary study. The items described in Table 2 allow us to characterize the most relevant aspects of the primary studies. This template allowed us to describe demographic data as well as descriptive data on barriers and facilitators.

To reduce bias, one author performed data extraction, and the second author verified the information extracted from each paper. Additionally, we classified primary studies based on the system views proposed by Krutchen [13]. These views are described as follows:Logical view: Describes the functionality that the system provides to end users. This represents what the system is supposed to do and the functions and services it offers.Development view: Details the system from a programming perspective and deals with the management of the software, i.e., it shows how the software system is divided into components and the dependencies between those components.Process view: Describes the processes in the system and how these processes communicate. In addition, from a system integrator’s perspective, it represents the step-by-step business and operational workflow of the components that make up the system.Physical view: Shows from a systems perspective all the physical components of the system as well as the physical connections between those components that integrate the solution (including services).

Because the logical view is related to the functionality of the system, we did not consider it for the classification of primary studies. Nevertheless, other views allow us to identify and describe the main barriers and facilitators based on different dimensions (development, process, and physical).

### 4.4. Replicability

We have created a repository [21] to detail the search protocol used in our study. The repository contains (i) the description of the search protocol, (ii) an archive of the primary studies and (iii) the images of our study.

## 5. Results

We obtained 61 primary studies as a result of our review (see Figure 2). We found primary studies conducted from 2008 to 2022 (see Figure 3). We did not find any primary studies in the years before 2008. The primary studies are mostly published in conferences and journals (see the list of selected primary studies in Table A1).

Figure 4 and Figure 5 illustrate the distribution of the primary studies with respect to views based on facilitators and barriers. On the facilitators side, Figure 4 depicts that 5% of the primary studies (3/61) mentioned facilitators in the process view, 33% (20/61) mentioned facilitators in the physical view and 49% (30/61) mentioned facilitators in the development view.

Regarding barriers, Figure 5 describes that 10% of the primary studies (6/61) mention facilitators in the process view, 30% (18/61) mentioned facilitators in the development view and 41% (25/61) mentioned facilitators in the physical view.

In the following sections, we describe the results obtained from each view, detailing the contributions of the studies focused on the facilitators and barriers detected.

### 5.1. Facilitators

Figure 6 illustrates the facilitators obtained in the primary studies. In the following sections, we describe each facilitator further.

#### 5.1.1. Development

One of the facilitators commonly mentioned in primary studies is the collaboration that exists between the development and design of AALSs. In this regard, studies mention that modern software development techniques make it possible to easily connect patient and community requirements in the design of AALSs. Because each community has a particular interest in AALSs, adaptable software design techniques to changing system requirements are highly relevant to the development and implementation of AALSs.

However, an important aspect mentioned by these studies is the variety of libraries and technologies available for the development, implementation, and deployment of AALSs. Most libraries point to programming languages such as Python and Java, which allow the use of specific packages and codes to process sensor data, alert signals, and patient movements within the home, among other qualities. Likewise, development teams have several technologies at their disposal that allow them to meet the primary needs of patients, their families, and health institutions. Some studies, such as S2–S25, mention that there are tools that allow the interoperability of data and functions through specific protocols.

Finally, another facilitator mentioned in the primary studies pointed to flexible software architectures. Many AALSs are designed with software architectures that allow the flexibility to add, modify, or remove functionalities requested by stakeholders. Currently, studies such as S44–S57 mention that there are software architecture styles that AALSs can use to address various requirements requested by stakeholders. For example, studies such as S33 and S43 used IoT to propose their respective solutions. The use of IoT implies designing a flexible, manageable, and sustainable system in the face of possible changes. These features can be addressed by using microservice architectures.

#### 5.1.2. Process

Communication among patients, families, and health institutions is essential for AALSs to be successful. Primary studies agree that two facilitators are important when defining processes, which point to the inclusion of both formal and informal communication networks and mutual assistance of the community. Regarding the first facilitator, primary studies point out that it is essential to consider that the success of AALSs strongly depends on the patient’s care network. This care network generally includes the following entities: family members, neighborhood, and health institutions. Since each entity has its own way of interacting with the AALSs, there are established processes and mechanisms capable of being integrated into the AALSs in order to consider all entities associated with the patient. Regarding the second, mutual assistance from the community is fundamental for AALSs. The studies describe that AALSs consider views and functionalities oriented to the community in order to integrate them into patient care. For this purpose, integration processes between the community and AALSs are defined to create policies and mechanisms that are effective before an adverse event related to a patient.

#### 5.1.3. Physical

Physical infrastructure presents facilitators relevant to AALSs’ success. One of the aspects mentioned by the primary studies is that there is a great variety of hardware that facilitates connection to medical devices. For example, smart watches, pacemakers, sensors, and night surveillance have an infrastructure capable of integrating different middleware to connect to AALSs.

Another relevant aspect manifested by primary studies is the growing trend of studies that propose low-cost sensors for implementing AALSs. Although private companies offer sensor services, the costs associated with them can be high. In some cases, the main users of AALSs are older adults, who do not have enough resources to pay for sensor services. This scenario generally occurs in developing countries as well. Therefore, studies such as S43 have proposed the development of low-cost sensors that provide the same (or almost the same) functionalities as more sophisticated sensors.

Since there is an increased interest in using AALSs, there has also been an effort to facilitate the installation of AALSs technology infrastructure. Some studies, such as S28–S32, emphasize that as medical devices become more automated and independent, the installation of technology in homes becomes increasingly expeditious. This situation is advantageous because other studies, such as S19 and S61, show that some patients do not accept the use of AALSs because they invade the privacy of their homes. Some medical devices require technology and infrastructure that must be installed in homes that do not accommodate all household members. Fortunately, with new connection technologies (such as 5G), the installation of cables has significantly reduced.

On the other hand, the research and creation of communication protocols between devices and systems have also positively impacted AALSs. Because many AALSs require fluid communication between different devices and networks, it is essential to have communication mechanisms and standards that define common communication modes. Several advances in communication protocols and networks have made it possible to define middleware layers that facilitate communication between medical devices and systems. This allows the interoperability of the AALSs to be managed, thus making the integration of new devices manageable and maintainable.

### 5.2. Barriers

Figure 7 shows the identified barriers. In the following sections, each barrier is described.

#### 5.2.1. Development

Primary studies report that the usability of AALSs is a barrier that hinders the use of this type of tool. Given that AALSs have a varied target audience ranging from clinical professionals to older adults, building graphical interfaces that are friendly, flexible, and anticipate potential errors is a challenge. Some studies report that older adults do not use AALSs despite their functionalities and benefits because of the difficulty in navigating mobile applications or web platforms.

Another important aspect to consider is the latency between the AALSs and databases. AALSs commonly have a layered architecture in which the data layer comprises several information sources. When medical sensors or devices generate a massive volume of information, some primary studies mention that this situation produces latency between the AALSs communication and the database, which eventually means that patient data are not real time. This implies that decision making can be affected by delays in information delivery.

There is no doubt that one of the main characteristics of AALSs is the integration of different medical devices and sensors. However, primary studies such as S16–S18 report that adverse events related to technical and configuration problems generally occur when installing such devices and sensors. For example, some devices and sensors do not have the correct range for transmitting data. This situation means that the services or software components responsible for receiving information from sensors and devices do not have information, which in turn means that AALSs cannot execute processes or alert patients, family members, or health institutions.

Software tests are processes in which the functionalities of a system are evaluated and analyzed in order to identify potential errors [22]. In general, primary studies indicate that the community poorly addresses software testing for AALSs. Although there are various software testing techniques, they generally focus on software components. Because of the above, there is not enough discussion on defining a stack of software tests focused only on AALSs. For instance, according to S56, software developers working with AALSs perform classic software tests. Although the above is an alternative, only some of the AALSs were tested.

Software development is based on the description of functional and extra-functional requirements (also known as non-functional requirements) [23]. Functional requirements describe the actions, tasks, and processes that software must satisfy. On the other hand, extra-functional requirements point to features, properties, business constraints, and others that the software must address. Complex requirements are composed of both types. However, those who define the requirements are mostly the stakeholders of a system. Regarding AALSs, the stakeholders of this type of system are generally patients, family members, and health institutions. That said, it is a complex task to gather all stakeholder needs and translate them into functional and extra-functional requirements. Each stakeholder wants something different from the AALSs; therefore, each needs to be addressed. However, there are complex processes (e.g., clinical processes) that make AALSs a complex system.

Although software components (e.g., middleware) allow the scalability of medical devices and sensors, the infrastructure and hardware do not necessarily support this. Some primary studies consider installing more connections and infrastructure in patients’ homes as a solution to this barrier; however, not all patients and relatives agree with these solutions because they invade the privacy and property of patients within their homes. Additionally, installing new infrastructure implies costs in essential goods (e.g., electricity) that must be financed by the patient.

Several primary studies have mentioned security as the main barrier to implementing and deploying AALSs. Recent studies such as [24] show that cyberattacks are becoming increasingly sophisticated, especially in emerging systems of high public interest such as AALSs. Given that AALSs are organized by different types of systems (Internet of Things, distributed systems, mobile applications, etc.), there are several ways in which this type of system can be compromised. This scenario implies that software developers must frequently implement techniques to create protection mechanisms for patient data and information contained in AALSs. Furthermore, the security problem is not only related to software development, but infrastructure and hardware are also compromised. On the other hand, the different sources that are connected to the AALSs must also be considered. As the number of sources increases, the potential risks that AALSs must address also increase.

#### 5.2.2. Process

The benefits of AALSs are diverse and mainly focus on the patients and their families. However, a common factor in most primary studies is that stakeholders do not fully understand when and where AALSs should be used. AALSs aim to provide ways to ensure that seniors in the home stay safe and can age in place. AALSs do as much as possible to make the patient’s aging possible at home. However, primary studies mention that some stakeholders do not understand the above and have incorrect expectations regarding what AALSs offer. These expectations range from believing that AALSs are structured like robots to thinking that AALSs are surveillance systems that violate patient privacy. On the other hand, some health institutions do not understand how to integrate AALSs into their clinical processes. Patient and emergency care processes are standardized in most health centers. For these reasons, it is difficult for these institutions to include AALSs in their processes.

Given that AALSs are gradually being accepted and implemented in health institutions, there is still uncertainty regarding the use of AALSs in clinical processes. It is important for health institutions to allocate resources correctly in the event of an emergency. Therefore, the alerts generated by AALSs must be honest.

#### 5.2.3. Physical

If AALSs require more processing or patient data, more sensors and infrastructure generally need to be installed. However, not all patients agree that their homes are filled with cables, devices, or systems that alter their privacy and normalcy. The main patients with AALSs are older adults or disabled people who are unaware of the use of technology. Therefore, if this technology disrupts the patient’s daily comfort and well-being, it is very likely that patients and their relatives will not accept AALSs.

One of AALSs’s main qualities of AALSs is that they can integrate different device and sensor ecosystems. However, primary studies have reported that managing these devices and sensors entails several difficulties. Primary studies focus on managing devices and sensors from different brands and companies. Each vendor offers its products, which in turn have data exchange formats. If AALSs have an ecosystem of devices and sensors from different vendors, managing the device data becomes an additional challenge.

This barrier is related to the cost of resources required to manage and manipulate the hardware necessary to use AALSs. As some sensors and medical devices are expensive, using AALSs is not always an alternative for low-income families and patients. In developing countries, the use of AALSs in homes is very precarious because the cost of managing and installing infrastructure and hardware exceeds the patient’s budget.

AALSs require a non-stop Internet connection to transmit the captured patient data. However, some studies, such as S1–S5, mention that some geographical regions in which older adults live make data transmission difficult. Data transmission has become a challenge in field or hill areas and compromises trust in the AALSs system. On the other hand, if the patient’s geographical area has signal difficulties, it is necessary to implement and install wi-fi enhancers, which eventually results in an additional cost.

Despite the benefits provided by AALSs, there is a factor that turns out to be relevant when implementing this type of system: energy consumption. Infrastructure, sensors, devices, networks, routers, and everything related to the operation of AALSs require additional energy consumption that is not generally included in the patient’s budget. The trend mentioned by some primary studies is that AALSs should be sustainable and green.

The reality of patient homes is not always adequate for installing AALSs. This type of system installation requires special connections, robust energy sources, and routers, among other devices. However, the correct conditions are not always sufficient to achieve proper installation. This means that if the services offered by the AALSs are not delivered entirely, the patient rejects the system’s installation.

## 6. Discussion

AALSs promote the creation of services that increase the time that people can live independently. In general, primary studies suggest that AALSs are aimed at primary end-users (the elderly) or caregivers, with the idea that they are extrapolated to other social groups, such as people with functional diversity. The hardware, software, and applications that generate AALSs must be able to collect data and monitor (and try to correct, if necessary) the life habits of the users.

Software developers have sufficient tools and technologies to develop and implement AALSs. The use of free libraries and technologies as well as flexible software architectures allows development teams to build AALSs systematically and efficiently. However, this is in contrast with the barriers identified in this study. Usability, scalability, availability, and security are the properties that AALSs must satisfy. These properties, in turn, are detected in the description of complex requirements. Because AALSs are generally distributed systems based on sensors and devices, the aforementioned properties are difficult to satisfy and evaluate in software testing. On the other hand, data access can become a complex issue if the algorithms implemented in AALSs rely on the data captured by the devices.

On the process side, AALSs are adaptable to integration into any process as long as the scope of the system is known. When the processes are well-defined in the AALSs, any additional services expected to be integrated will be able to do so effectively. Nevertheless, if the context of AALSs is not well defined, confidence in using AALSs to improve the quality of life of patients may be diminished.

There is no doubt that for AALSs to operate properly, physical infrastructure is required. Primary studies describe diverse facilitators that enable the physical infrastructure of AALSs to be accessible and non-invasive for patients. Several primary studies have provided various contributions, such as low-cost sensors, data communication protocols, and expedited physical installations. However, the results of our study reveal that there are still many challenges that need to be addressed. These challenges can be categorized as technical and social. On the one hand, some challenges relate to the technical management of AALSs (devices, hardware cost, and others), and on the other hand, some challenges involve patients. These challenges mainly point to how patients tolerate and accept technological invasions at home.

Overall, the results of our study showed that there are more barriers to AALSs than facilitators. It is important to note that the detected facilitators are relevant to AALSs because they allow them to become increasingly attractive to patients. However, the identified barriers may be the inspiration for addressing the challenges presented by AALSs. Given the impact and great potential of AALSs in improving patients’ quality of life, the barriers described in Figure 7 can be used as a starting point for emerging research.

### 6.1. Implications for Researchers

Although our study revealed pragmatic findings on the facilitators related to the development and deployment of AALS, the barriers illustrate interesting research challenges. From a development point of view, the identified barriers mainly point to quality attributes and non-functional requirements. Concerning usability, availability, scalability, and security, these quality attributes present challenges related to satisfying these attributes in AALSs. Each of the quality attributes mentioned above aims to satisfy specific properties in AALSs; however, these properties must be consistent with the needs of users and patients. Some of these properties, such as security and availability, involve multidimensional research challenges, ranging from how to design the software architecture of the AALS to what kind of infrastructure is required to satisfy the needs of stakeholders with respect to these quality attributes. On the process side, research challenges are directed toward how an AALS is understood by patients and health-care institutions. Some primary studies have reported that many patients, families, and even health institutions need to help understand what AALSs provide. In this scenario, dissemination and implementation strategies in health care [25] can be an alternative for stakeholders to understand the impact of AALSs on patient care. Finally, from a physical point of view, challenges are oriented toward technological infrastructure. The development of low-cost and friendly medical devices, flexible infrastructure, and energy consumption are potential sources of research that entail multidimensional work between technology, patients, and health-care institutions.

### 6.2. Implications for Practitioners

The results of this study are relevant for practitioners when developing and implementing AALS. Regarding the facilitators, the results described in Figure 6 can be used as guidelines or recommendations for practitioners to consider when making decisions. From a development point of view, practitioners can use the results to evaluate which software architecture is the most suitable for designing an AALS, for example. The facilitators described in the process view allow practitioners to identify and characterize the main stakeholders of AALSs. Finally, from a physical point of view, the results of our study highlight technological infrastructure aspects that practitioners should consider when deploying AALSs. Given the significant increase of technical solutions in the market for system development, the facilitators described in Figure 6 can be utilized to make strategic decisions toward a successful design and deployment of AALSs.

On the contrary, the barriers depicted in Figure 7 act as potential risks that practitioners should consider. The barriers described in the development, process and physical views allow practitioners to anticipate potential concerns that may arise when developing and deploying AALSs. In turn, Figure 7 can be used to anticipate adverse scenarios that may arise in the development and deployment of AALSs. These scenarios allow practitioners to analyze the artifacts and archetypes that should be considered to address potential problems that will emerge in the development and deployment of AALSs.

## 7. Limitations

We analyze the limitations of our study through threats to validity. In this regard, we use the classification of Wholin et al. [20], which describes approaches to classifying and mitigating threats.

Threats to internal validity point to elements that may affect study results. We identified that study selection might be associated with bias in terms of article search. To mitigate this threat, we based our study on a review process widely used by the clinical community [12]. Additionally, we included inclusion and exclusion criteria to select the primary studies. Furthermore, we conducted several cross-validations with external collaborators to validate and evaluate each primary study.

Threats to external validity focus on conditions that limit the generalizability of results. In this regard, the main threat we detected is whether the primary studies represent the gains and pains of AALS. To mitigate this threat, we invited health professionals and technology experts who collaborated with our study to discuss and analyze each primary study in order to obtain feedback. Additionally, we used our previous experience [26] in AALS to identify potential gains and pains.

About threats to construct validity, these threats are related to the generalizability of the results to the concept associated with the SLR. The main threat is the subjectivity of our results. To mitigate this threat, we iterated the main steps of the SLR with a group of collaborators and compared the results independently.

## 8. Conclusions

This paper describes the results of a systematic literature review that identifies barriers and facilitators to using AALSs. A total of 61 primary studies obtained through a search protocol were reviewed and classified into three views: development, process and physical infrastructure. Regarding development, we identified three facilitators and seven barriers mentioned in the primary studies. On the process side, we describe two facilitators and barriers. Finally, we identified four facilitators and six barriers on the physical infrastructure side. The results obtained in this study reveal that there are indeed facilitators that enable the development and deployment of an AALS to be effective. On the other hand, the barriers identified in the study reveal several opportunities to address emerging challenges in AALSs related to improving the well-being of older adults and disabled people.

To further our research, we plan to expand our study by analyzing AALSs in production. We will extend the findings with practical evidence in order to describe guidelines for practitioners to follow when developing and installing AALS in homes.

## Figures and Tables

**Figure 1 ijerph-20-05020-f001:**
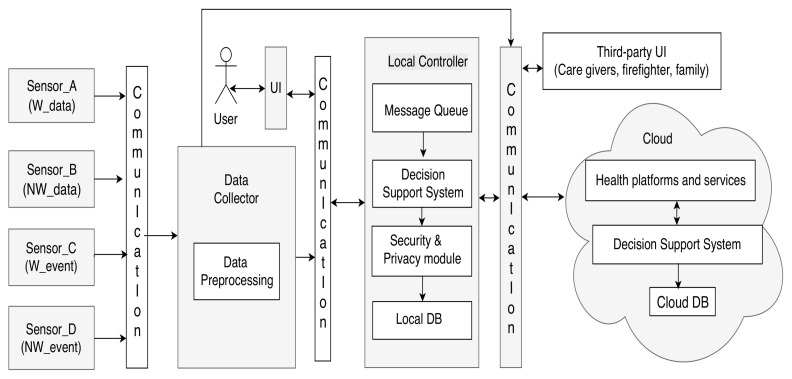
Architecture of an AALS described in [15].

**Figure 2 ijerph-20-05020-f002:**
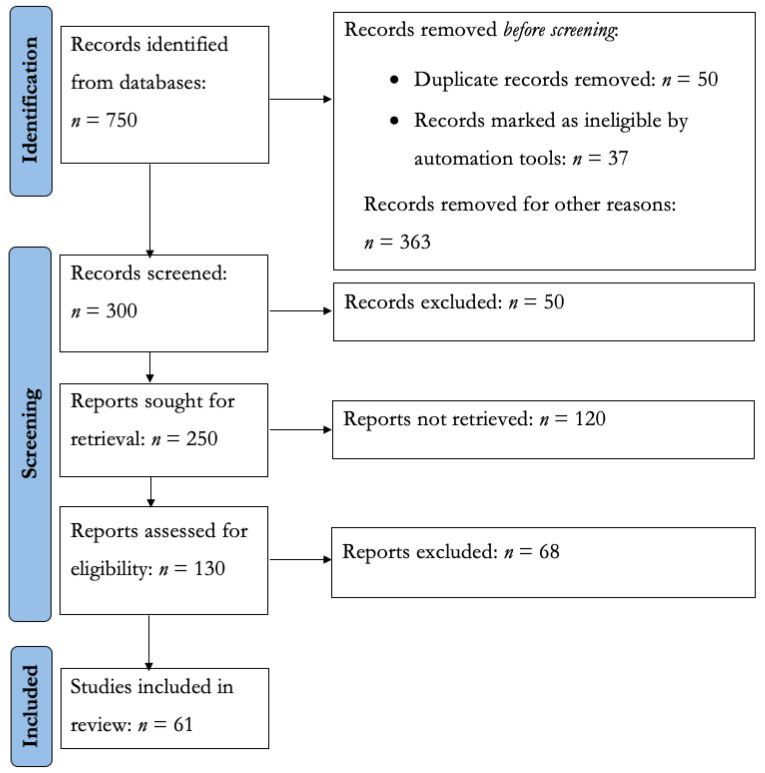
PRISMA flowchart of primary studies selection.

**Figure 3 ijerph-20-05020-f003:**
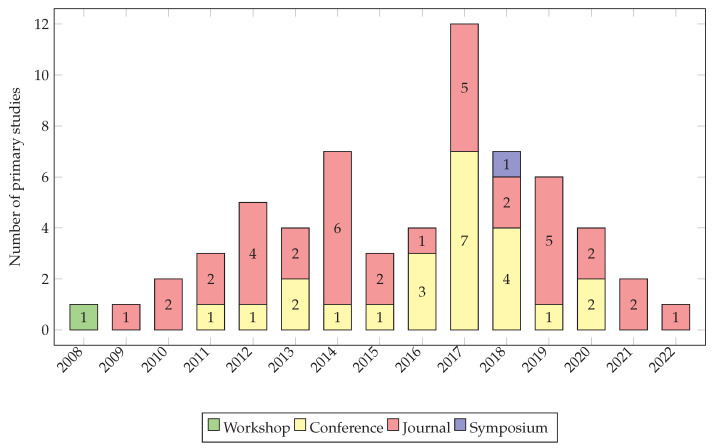
Distribution of publication type.

**Figure 4 ijerph-20-05020-f004:**
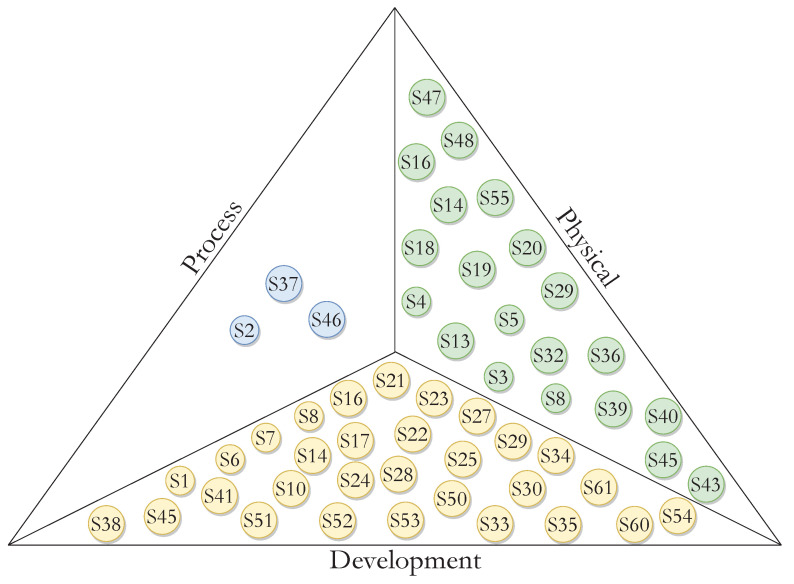
Distribution of primary studies concerning facilitators. Blue circles correspond to process, green to physical, and yellow to development.

**Figure 5 ijerph-20-05020-f005:**
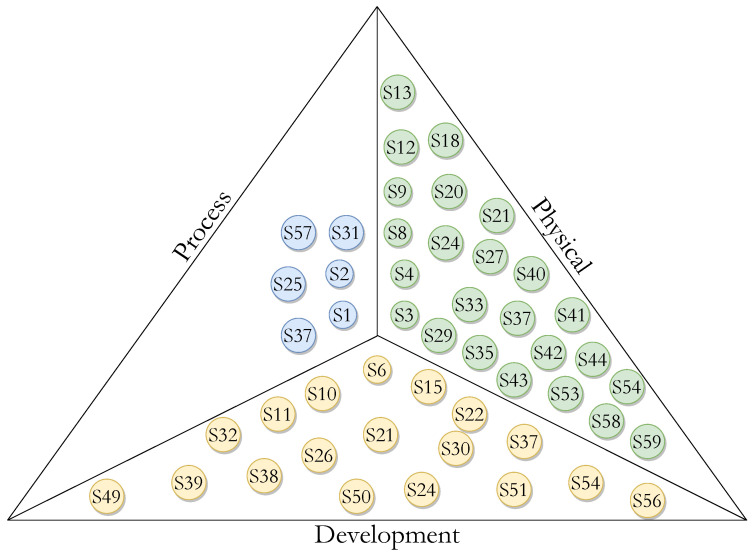
Distribution of primary studies concerning barriers. Blue circles correspond to process, green to physical, and yellow to development.

**Figure 6 ijerph-20-05020-f006:**
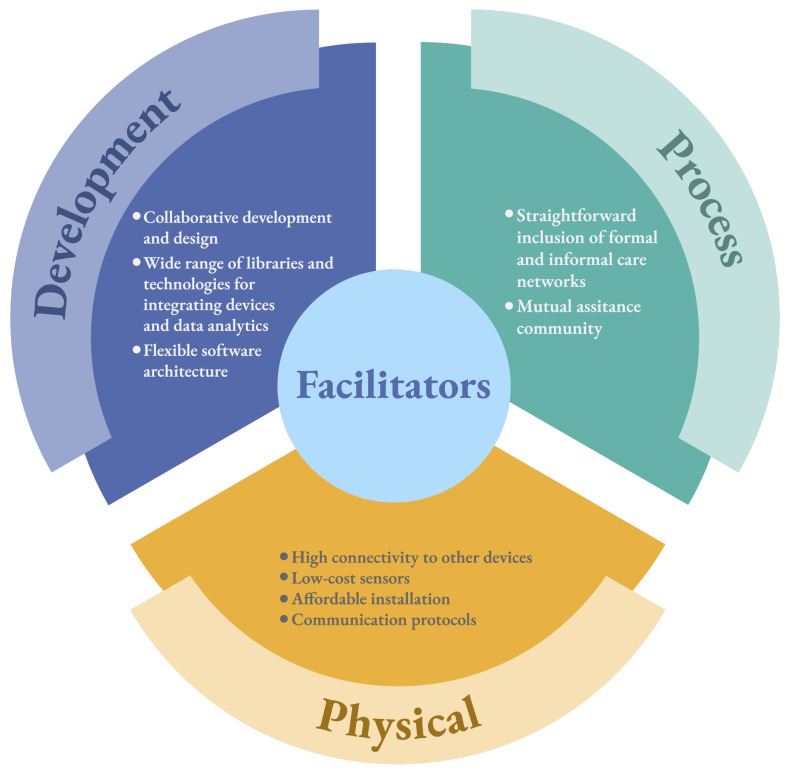
Description of the detected facilitators classified by the system views.

**Figure 7 ijerph-20-05020-f007:**
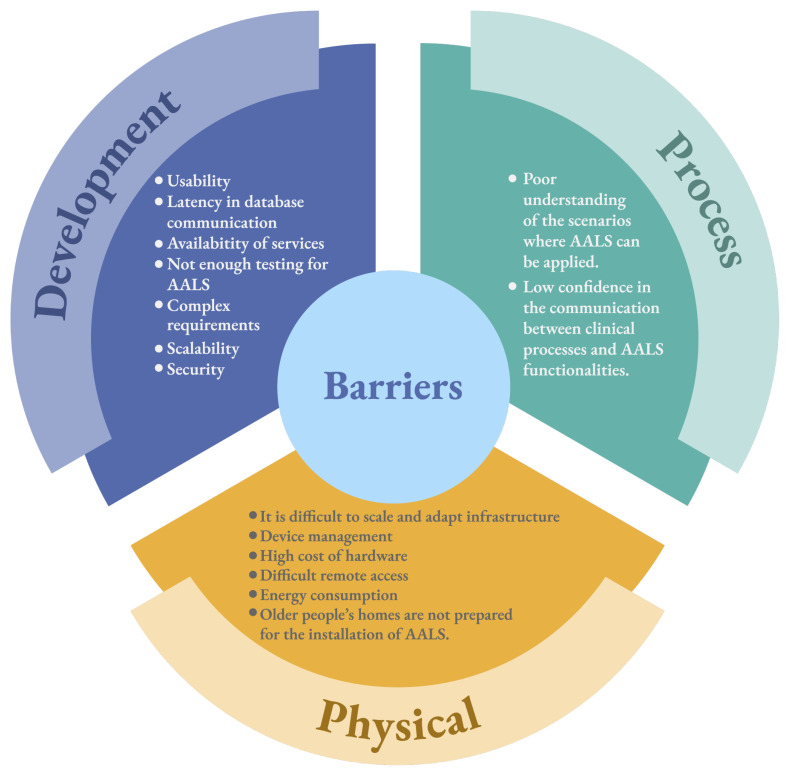
Description of the detected barriers classified by the system views.

**Table 1 ijerph-20-05020-t001:** Description of the search string.

Approach	Description	Keywords
Population	Papers related to AALSs and their derivatives	“ambient-assisted” OR “ambient assisted” OR “living”
	Paper related to the types of technologies that support AALSs	“system*” OR “platform” OR “application” OR “software” OR “technology”
Intervention	Papers related to health and care	“health” OR “care” OR “health care”
Output	Data regarding facilitators and barriers	“barrier” OR “disadvantage” OR “pitfall” OR “pain” OR “facilitator” OR “advantage” OR “gain” OR “benefit”

**Table 2 ijerph-20-05020-t002:** Description of the template used to characterize primary studies.

Data Item	Description
ID	This item describes a specific identification for each primary study
Name	Name of the study
Authors(s)	List of authors of the study
Year	Year of study publication
Venue	Description of where the study was published or presented
Facilitator	List of the facilitators described in the study
Barrier	List of barriers described in the study

## Data Availability

The search protocol presented in this study are openly available in [21] at https://doi.org/10.5281/zenodo.7578848 (accessed on February 2023).

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
