# Peer review of "Barriers and Facilitators of Ambient Assisted Living Systems: A Systematic Literature Review"

_ijerph, 2023, doi:10.3390/ijerph20065020_

Round 1

Reviewer 1 Report (Previous Reviewer 2)

Dear authors,
I appreciate that you have made a number of corrections and improvements to the manuscript according to the reviewers' comments. If I focus only on my comments, we did not understand each other in the case of my first comment.

I did not want to add the mentioned chapter/book to your review (by referring to the given chapter, I just wanted to emphasize that this is an important area for ALLSs). I wanted to add some discussion or reasons why the areas relevant for all types of disabled people (in this case, I don't just mean older people, but also younger people with specific disabilities for who ALLSs are also developed) are not mentioned in your review? Why are there no studies on barriers and facilitators related to ALLSs e.g. for people with blindness or deaf people? I welcome your reflection on this problem in your manuscript.

I don't have any new comments.

Author Response

Comment 1: I did not want to add the mentioned chapter/book to your review (by referring to the given chapter, I just wanted to emphasize that this is an important area for ALLSs). I wanted to add some discussion or reasons why the areas relevant for all types of disabled people (in this case, I don't just mean older people, but also younger people with specific disabilities for who ALLSs are also developed) are not mentioned in your review? Why are there no studies on barriers and facilitators related to ALLSs e.g. for people with blindness or deaf people? I welcome your reflection on this problem in your manuscript.

Reply to comment 1: We thank the reviewer for this comment. We fully agree with the reviewer regarding the potential barriers and facilitators of AALS in people with reduced capacity. Moreover, we have no doubt that the reviewer’s point is very interesting to explore. However, our study did not aim at barriers and facilitators from the patient's point of view; it aimed to AALSs in operational settings. In our previous experience, we realized that AALSs have advantages and disadvantages similar to any type of IT system; however, we realized that there is very little pragmatic evidence on the implementation and deployment of AALSs. The point of view mentioned by the reviewer is very interesting and has the potential for further research. However, as mentioned, our study aims at a more technical discussion of the facilitators and barriers of AALSs, as reported by patients, families, and clinicians. Additionally, as mentioned in the article, our aim is to make clinicians and practitioners aware of the operational barriers and facilitators of AALS, as well as other facilitators and barriers that may emerge in other domains, such as patients with reduced capacity.

Reviewer 2 Report (New Reviewer)

This paper presents a systematic literature review on barriers and facilitators for using ambient-assisted living systems.

Suggestions and questions:

1. Methods could be better described in the abstract.

2. In the paper, the motivation for AALSs is the [remote] monitoring of older adult disabled people. Is AALSs developed only for such individuals?

3. Related studies (i.e., relates reviews) should be presented to mae evident the novelty of this review.

4. Consider "Additionally, we have created a repository [14] to detail the search protocol used in our study. The repository contains (i) the description of the search protocol, (ii) an archive of the primary studies and (iii) the images of our study."; this text should be in the methodology section.

5. Moreover, why was [14] not presented as supplementary material? It could be presented there.

6. Figure 2 should not be part of the methods, it is a result.

7. Consider "...based on three steps: identification, screening and included"; 'included' is not a step.

8. Consider "The identification process was conducted between January and October 2022."; when were the searches carried out in the article databases? 

9. Related work section is badly positioned. It could be the third section.

10. Important point: I think the study objective is not well declared. Consider "This paper describes the facilitators and barriers of using AALSs in preventing and caring for older adults"; There is no particular result or discussion focused on 'older adults'.

Specific points:

- The subheading (i.e., the name) '3.3. Included' does not make sense. It presents the data extraction process.

- Figures 6 and 7 are too big.

Author Response

Comment 1: Methods could be better described in the abstract.

Reply to comment 1: Thank you very much for your comment. We have described the method in the abstract for better understanding.

Comment 2: In the paper, the motivation for AALSs is the [remote] monitoring of older adult disabled people. Is AALSs developed only for such individuals?

Reply to comment 2: Thank you for your comment. Indeed, AALSs are not only developed for older adults, they are also developed for people with reduced capacity. We have updated the text of the manuscript to describe this.

Comment 3: Related studies (i.e., relates reviews) should be presented to mae evident the novelty of this review.

Reply to comment 3: Thank you very much for your comment. The manuscript has a section where related work is discussed, but we have placed it at the end of the manuscript. Taking into consideration the reviewer's comment, we have moved the section to the beginning of the manuscript.

Comment 4: Consider "Additionally, we have created a repository [14] to detail the search protocol used in our study. The repository contains (i) the description of the search protocol, (ii) an archive of the primary studies and (iii) the images of our study."; this text should be in the methodology section.

Reply to comment 4: Thank you very much for your comment. We have created a new section 4.4 to describe the study protocol to take the text out of the introduction.

Comment 5: Moreover, why was [14] not presented as supplementary material? It could be presented there.

Reply to comment 5: Thank you very much for your comment. We have added the repository in the data availability section.

Comment 6: Figure 2 should not be part of the methods, it is a result.

Reply to comment 6: Thank you for the suggestion. We agree that the figure should be in the results section. Therefore, we have moved the figure to section 5 of the manuscript.

Comment 7: Consider "...based on three steps: identification, screening and included"; 'included' is not a step.

Reply to comment 7: Thank you for your comment. We agree that "included" does not sound like a step. However, the definition of the PRISMA methodology considers the step called "included". For instance, here are some papers that use the same name "included" for the step in the methodology:

[1] Cortés-Denia, D., Lopez-Zafra, E., & Pulido-Martos, M. (2021). Physical and psychological health relations to engagement and vigor at work: A PRISMA-compliant systematic review. Current Psychology, 1-16. (url: https://link.springer.com/article/10.1007/s12144-021-01450-y#Sec3)

[2] Park, J. H., Jung, S. E., Ha, D. J., Lee, B., Kim, M. S., Sim, K. L., ... & Kwon, C. Y. (2022). The effectiveness of e-healthcare interventions for mental health of nurses: A PRISMA-compliant systematic review of randomized controlled trials. Medicine, 101(25), e29125-e29125. (url: https://www.ingentaconnect.com/content/wk/medi/2022/00000101/00000025/art00003?crawler=true&mimetype=application/pdf)

[3] Vakharia, V. N., Khan, S., Marathe, K., Giannis, T., Webber, L., & Choi, D. (2021). Printing in a Pandemic: 3D printing solutions for healthcare during COVID-19. A Protocol for a PRISMA systematic review. Annals of 3D Printed Medicine, 2, 100015. (url: https://www.sciencedirect.com/science/article/pii/S2666964121000102)

Comment 8: Consider "The identification process was conducted between January and October 2022."; when were the searches carried out in the article databases?

Reply to comment 8: Thank you for your comment. We have decided to mention a time interval to describe the search for primary studies as the research team during this range of months reviewed the databases described in the manuscript. Since each researcher and collaborator executed the search for primary studies at different times, we are not precise about when each database was reviewed. At the end of October, we consolidated all the papers we collected and proceeded to follow the PRISMA methodology.

Comment 9: Related work section is badly positioned. It could be the third section.

Reply to comment 9: Thank you for your comment. We have restructured the manuscript by moving the related work section to the beginning (section 3).

Comment 10: Important point: I think the study objective is not well declared. Consider "This paper describes the facilitators and barriers of using AALSs in preventing and caring for older adults"; There is no particular result or discussion focused on 'older adults'.

Reply to comment 10: revisar cunado vea el ingles

Comment 11: The subheading (i.e., the name) '3.3. Included' does not make sense. It presents the data extraction process.

Reply to comment 11: Thank you for your comment. In this section we describe what methodology we use to include primary studies in our study. The aim of this step, according to the PRISMA methodology, is to describe how the researchers will organize and detail the papers to be included in the review. Thanks to the methodology we have proposed in section 4.3, we have classified and categorized the primary studies we used in our study. Moreover, the methodology we used helped us to detect 7 papers that, despite passing the screening step, did not meet the research objectives, as these papers were oriented in other technical contexts of AALSs (public and private health systems).

Translated with www.DeepL.com/Translator (free version)

Comment 12: Figures 6 and 7 are too big.

Reply to comment 12: Thank you for your comment. We have reduced the size of the images.

Round 2

Reviewer 2 Report (New Reviewer)

The authors improved the paper based on my comments. Most of my questions/comments were addressed.

This manuscript is a resubmission of an earlier submission. The following is a list of the peer review reports and author responses from that submission.

Round 1

Reviewer 1 Report

The methodology used for the review is sound. The content of the article provides a very useful collection of papers pointing to the practical aspects of implementing AALS, and a succinct description of the barriers and facilitators for the implementation of AALS covered in the referenced papers. 

I gave three stars to the section "Are there appropriate and adequate references to related and previous work" because although the authors cover an extensive number of papers on the subject of AALS, do not include references to other papers providing literature reviews in AALS. 

I recommend the publication of the article. 

Reviewer 2 Report

I appreciate the chance to serve as a reviewer on this review paper. The paper is interesting for IJERPH. I recommend accepting the paper after significant revisions.

The paper is well structured, and the systematic review is adequately described. All approaches and perspectives are explained. On the other hand, the paper lacks a deeper discussion (the paper does not go into detail), a summary of the main implications (for science/practice) and suggestions for future research in AALSs.

I hope that my comments help you to improve your paper.

Major comments:

1) "Ambient Assisted Living Systems (AALSs) are solutions that use information and communication technologies to support the care of the growing population of older adults." Ambient Assisted Living Systems are not only about older people, but it also covers disabled people (it is an important area in AALSs).
You completely missed these publications; you need to explain why?
See, e.g. book: https://www.taylorfrancis.com/books/edit/10.1201/b18520/ambient-assisted-living-joel-jose-rodrigues-nuno-garcia chapter 15 AAL Applications to Specific Areas.

2) Add suggestions for areas of further research that need to be solved by researchers all over the world in terms of the given perspectives, as well as problems that need to be solved in the coming years. Ideally, in a table with links to articles used in the review study.

Minor comments:

+ Page 3, with figures, research questions, and the PRISMA diagram, is confusing for the reader. It is only a formal aspect, but don't forget to check it after the acceptance and DTP phases.

+ Inappropriate statement, rephrase: "Therefore, any change that implies an alteration in your daily comfort and well-being is a rejection of the AALSs."

Reviewer 3 Report

The authors report the results of conducting a systematic literature review to investigate barriers and facilitators in AAL systems.

The work is relevant to the domain and is well organized.

However, this reviewer identified the following issues.

The main issue of this work is the nature of this study. The authors said they conducted a systematic literature review (SLR), however, the review is more superficial and less rigorous for an SLR. For instance, in an SLR quality assessment is required, which was not conducted in this work. 

In Table 1, the population description is not clear. The keywords related to population are not adequate, for instance, how software, platform, technology, or application (all of them generic terms) are a synonym of AAL?

Also, in the identification stage, it is required the authors explain the timeline required for planning, conducting, and reporting the study. Also, how many researchers participated in the works screening? 

Additionally, it is required to explain which data were extracted from each study to answer the research questions. 

In the execution of the search string the authors must be clear about possible searching filters (e.g., year, type of venue, title, abstract, keywords, etc).

This reviewer used the Scopus data library to validate the string and found more than 470.000 registers in such a base. The search string used for this intention was:  

Therefore, the identification stage is not transparent and does not allow the replication and verification of this study. 

This work's goal is to identify barriers and facilitators for AAL systems as designed in the research questions. However, the authors classified papers as architecture layers, based on 4+1 views model. What is the rationale for this decision? 

The results section lacks transparency in reporting the results. For instance, which studies reported facilitators? and benefits? Which studies reported each type of facilitator and benefit? What was the facilitator or benefit most highlighted by the studies? The results analysis is very limited and did not offer enough evidence to believe in the statements exposed in such a section.

Additional suggestions:

- Figures 1 and 2 are mixed with the research questions, making the reading difficult. Let figures close to the text related to them.

- The manuscript must be proofread. Some grammatical errors were found.